# A machine learning approach to predict extreme inactivity in COPD patients using non-activity-related clinical data

Bernard Aguilaniu[1]*, David Hess[2], Eric Kelkel[3], Amandine Briault[4], Marie Destors[4], Jacques Boutros[5], Pei Zhi Li[6], Anestis Antoniadis[7,8]

**1** Faculty of Medicine and Pharmacy, Grenoble Alps University, Grenoble, La Tronche, France, **2** Colibri-Pneumo Program, Association for Consolidation of Knowledge and Practices of Pulmonology, Grenoble, France, **3** Centre Hospitalier Metropole Savoie, Chambery, France, **4** CHU Grenoble Alpes, La Tronche, France, **5** Department of Pulmonary Medicine and Oncology, CHU de Nice, FHU OncoAge, Université Côte d'Azur, Nice, France, **6** Respiratory Epidemiology and Clinical Research Unit, McGill University, Montreal, QC, Canada, **7** Jean Kuntzmann Laboratory, Grenoble, France, **8** Department of Statistical Sciences, University of Cape Town, Rondebosch, Cape Town, Western Cape, South Africa

* b.aguilaniu@gmail.com

**Data Availability Statement:** French law requires that patients be informed of the analyses that are performed on their data, even if they are anonymized. Patients have the right to refuse

## Abstract

Facilitating the identification of extreme inactivity (EI) has the potential to improve morbidity and mortality in COPD patients. Apart from patients with obvious EI, the identification of a such behavior during a real-life consultation is unreliable. We therefore describe a machine learning algorithm to screen for EI, as actimetry measurements are difficult to implement. Complete datasets for 1409 COPD patients were obtained from COLIBRI-COPD, a database of clinicopathological data submitted by French pulmonologists. Patient- and pulmonologist-reported estimates of PA quantity (daily walking time) and intensity (domestic, recreational, or fitness-directed) were first used to assign patients to one of four PA groups (extremely inactive [EI], overtly active [OA], intermediate [INT], inconclusive [INC]). The algorithm was developed by (i) using data from 80% of patients in the EI and OA groups to identify 'phenotype signatures' of non-PA-related clinical variables most closely associated with EI or OA; (ii) testing its predictive validity using data from the remaining 20% of EI and OA patients; and (iii) applying the algorithm to identify EI patients in the INT and INC groups. The algorithm's overall error for predicting EI status among EI and OA patients was 13.7%, with an area under the receiver operating characteristic curve of 0.84 (95% confidence intervals: 0.75–0.92). Of the 577 patients in the INT/INC groups, 306 (53%) were reclassified as EI by the algorithm. Patient- and physician- reported estimation may underestimate EI in a large proportion of COPD patients. This algorithm may assist physicians in identifying patients in urgent need of interventions to promote PA.

analyses based on their personal data. In our case, therefore, there are legal restrictions on the public sharing of data. To access the data, it is necessary to contact Colibri directly (contact@colibri-pneumo.fr) which is the institution that holds the data.

**Funding:** The COLIBRI web consultation platform is supported by contractual partnerships with Agir à Dom, AstraZeneca, Boehringer Ingelheim, Chiesi, GlaxoSmithKline and Novartis. BA, DH, and AA received grants from Agir à Dom, AstraZeneca, Boehringer Ingelheim, Chiesi, GlaxoSmithKline, and Novartis for the conduct of the study. The funders had no role in study design, data collection and analysis, decision to publish, or preparation of the manuscript.

**Competing interests:** BA, DH and AA received grants from Agir à Dom, AstraZeneca, Boehringer Ingelheim, Chiesi, GlaxoSmithKline and Novartis for the conductif of the study. This does not alter their adherence to PLOS ONE policies on sharing data and materials.

## Introduction

Patients with chronic obstructive pulmonary disease (COPD) are known to be substantially less physically active than age- and sex-matched healthy subjects [1]. Several studies have shown that low physical activity (PA) levels are associated with poor prognosis in COPD patients [2, 3], yet pulmonary rehabilitation programs that incorporate endurance and strength training have shown significant benefit in this patient population [4]. Thus, accurate identification of the true PA status is a crucial factor in ensuring that the least active patients, who would be expected to derive the greatest benefit from PA, can be encouraged to become more active and/or referred to a rehabilitation program.

Several methods have been devised to assess and quantify PA levels in patients with various respiratory diseases. In particular, accelerometers can be worn over several days to analyze the full range of different activities and their distribution over time. Data from such devices have generally correlated well with assessments of daily metabolic expenditure, as measured using the doubly labeled water method, and accelerometers are also sufficiently sensitive to detect low levels of PA in COPD patients [4]. These quantitative studies have estimated that approximately 26%–30% of COPD patients are physically inactive and exhibit sedentary behavior, both of which are independently associated with an increased risk of morbidity and mortality [3, 5, 6]. However, accelerometry requires considerable cost, time, and effort commitments on the part of the patient and physician, and it is generally considered impractical for routine clinical use. At the same time, clinical interviews and patient questionnaires alone cannot accurately determine the patient's true PA level [7]. To improve this situation, the PROactive consortium proposed that a combination of questionnaires and accelerometric measurements be used to assess the behavior of COPD patients [8, 9]. Nevertheless, this approach does not eliminate the drawbacks of accelerometry, and therefore does not resolve the primary clinical concern, which is to accurately and objectively detect extreme inactivity (referred to hereafter as EI) in patients whose PA status initially presents as unclear or equivocal [10, 11]. Although such patients may be identified during consultation with experienced practitioners, it is likely that a significant percentage of EI patients fall under the radar of clinical vigilance, which most often focuses on respiratory function. Given the proven benefit of pulmonary exercise programs in COPD patients, we therefore sought to develop a predictive algorithm that can reliably detect EI patients, who might most benefit from interventions such as pulmonary rehabilitation programs.

We hypothesized that certain physiological and clinical variables may be more frequently observed (through cause or effect) among patients at the extreme ends of the PA spectrum (i.e., EI and overtly active [OA] patients), and that such 'phenotype signatures' composed of non-PA-related variables could be used to develop the predictive algorithm.

## Materials and methods

### Patients and data collection

This was a retrospective analysis of data submitted to the COLIBRI-COPD database [12, 13], which has been authorized by the French national commission on personal data privacy (Commission Nationale de l'Informatique et des Libertés, CNIL, #2013–526). The requirement for written consent was waived in this observational study in accordance with French law. Patients provided oral informed consent to their physician. At the time of the analysis, data were available from 5035 initial consultations for COPD patients (Fig 1). We selected 1409 patients with comprehensive information on 22 specific variables (see Table 2) in the areas of anthropometry, smoking habits, resting pulmonary function, comorbidities, exacerbations during the preceding year, Global Initiative for COPD (GOLD) ABCD classification, and self-

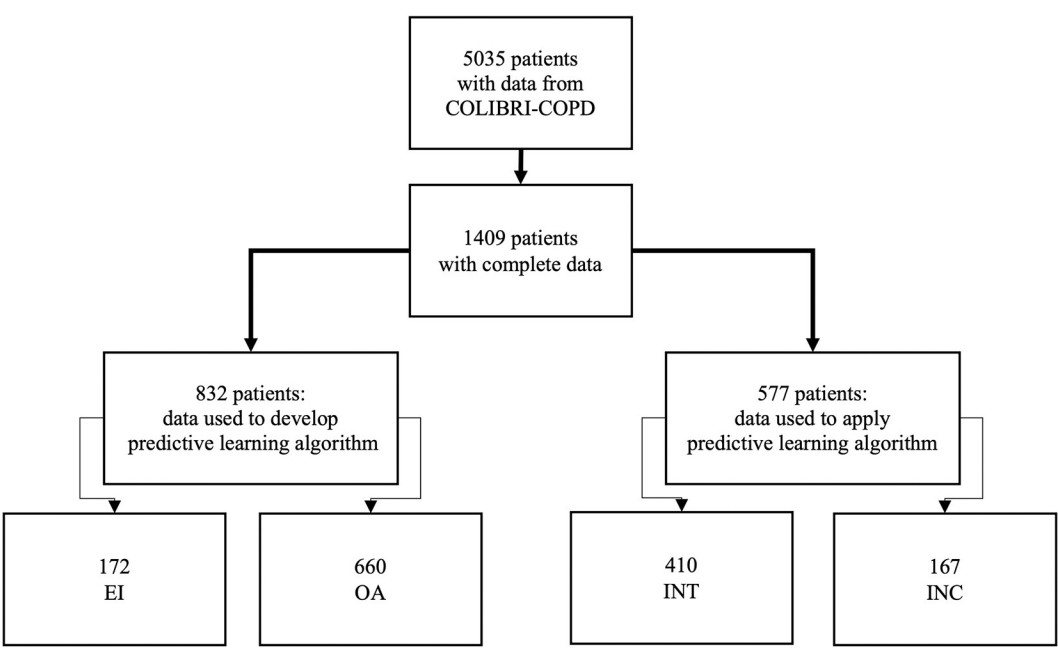

**Fig 1. Study design.** See Table 1 for definitions of activity categories.

reported questionnaires: the modified Medical Research Council dyspnea scale (mMRC) [14] and Disability related to COPD Tool (DIRECT), both of which assess dyspnea [15, 16]; the COPD Assessment Test (CAT), which assesses quality of life [17]; and the Hospital Anxiety and Depression Scale, which separately assesses anxiety and depression [18, 19].

## Construct of the predictive machine learning

We first categorized a cohort of COPD patients into one of four activity levels based on the patient's own estimates of their PA (daily walking time) and the physician's estimates of the patient's PA intensity level (domestic, recreational, and fitness-directed). We then tested existing machine learning processes already in use for predicting disease outcomes using routine clinical data [20, 21], and trained the model to identify an EI signature using clinicopathological data from a subset (80%) of patients in the EI and OA categories. After training, we tested the algorithm's predictive validity on the remaining 20% of patients in the EI and OA categories, and then evaluated its ability to detect EI patients in the intermediate (INT) or inconclusively determined (INC) PA categories.

## Definition of PA categories

Assignment of patients to PA categories was based on physician estimates of the predominant intensity level of the patient's daily PA: domestic (D, in-home activities), recreational (R, mostly outside the home), or active (A, devoted to maintaining physical fitness) and patient estimates of the average daily walking time outside the home (including weekends): <10 min, 10–30 min, 30–60 min, and >60 min. Based on these criteria, we constructed a 3 × 4 table to identify four main PA categories: (i) least active (EI, n = 172); (ii) most active (OA, n = 660); (iii) intermediate activity level (INT, n = 410), which had three subcategories (a, b, and c); and (iv) incompatible (INC, n = 167), which had four subcategories (a, b, c, and d) and consisted of patients whose self-reported and physician-reported activities were considered conflicting

(Table 1). Descriptive clinical and functional characteristics of COPD patients stratified by PA categories are presented as mean ± standard deviation. Comparisons between PA categories were performed by Kruskal-Wallis tests and ANOVA with ordinal factors test (ordAOV).

## Predictive statistical methods

The predictive machine learning method was developed in five steps. (i) We first verified that the EI variable and its variability correlated well with a set of continuous and categorical variables. Then, we performed an explanatory canonical discriminant analysis of mixed data followed by a scree plot to select the statistically significant canonical variables to be used in more elaborate individual predictive models. After this step, a reduced rank display (S1 Fig) showed that two canonical discriminant projections accounted for 98.6% of the variation between categories, of which 95.8% concerned EI and OA, while the projection of INT and INC on the two canonical directions was very slight. (ii) Based on this, we opted to develop an algorithm focused on individual prediction of the two most extreme categories; EI (n = 172) *versus* OA (n = 660). The predictive model was developed using an ensemble regression and classification algorithm [22] with a version for balancing error in unbalanced data (weighted random forest, WRF). To account for random effects, such as the physician identity or study center, we also combined the random forest methodology with generalized linear mixed models using the binary mixed model (BiMM) forest algorithm [23]. (iii) Data from the 832 patients in the EI and OA groups were randomly selected; of these, we used data from 666 patients (80%) to develop the model and data from the remaining 20% (166 patients) to assess its accuracy (i.e., predictive error). (iv) In the next step, we addressed the imbalance in our final prediction using a recent hyper-ensemble of SMOTE under sampled random forests (HyperSMURF) method, which is based on resampling techniques and a hyper-ensemble approach (S2 Fig). (v) Finally, once validated, the algorithm was applied to patients in the combined INT and INC subcategories.

Descriptive results are presented as mean ± standard deviation. The performance of the algorithm for predicting EI and OA is expressed as overall error, weighted accuracy, true negative value, true positive value, and sensitivity. Additional performance measurements included area under the precision and recall curve (AUPRC) and area under the receiver operating characteristic curve (AUROC).

## Results

### Descriptive results

Table 1 shows the distribution of the 1409 patients into four categories and 12 subcategories according to the combination of patient and physician estimates. The reference category EI (n = 172) was composed of patients with the lowest duration and intensity PA level (subcategory D and <10 min walking/day), whereas the OA category (n = 660) included the most active patients (subcategory R or A and >30 min walking/day). Patients who spent short times (≤30 min) in daily activities were referred to as the INT group (n = 410) and were subcategorized as a, b, or c, depending on the physicians' estimate of the activity intensity (Table 1). Finally, patients whose self- and physician-reported subcategories were incompatible were referred to as the INC group (n = 167) and were further assigned to a, b, c, or d groups based on the time and intensity. The seven categories encompassed by INC a–d and INT a–c together account for about 40% of the total cohort, highlighting the need for a tool to more accurately assess daily PA.

After validation and predictive validity testing (see next section), we applied the algorithm to patients in the full cohort as well as the INC and INT categories and determined the number

**Table 1. Categorization of physical activity levels in COPD patients according to combined patient- and physician-derived estimates.**

| Patient's Estimate (daily walking time; n = 1409) | Physician's Estimate (activity intensity; n = 1409) | | |
|---|---|---|---|
| | (D)omestic | (R)ecreational | (A)ctive |
| | n = 504 | n = 530 | n = 375 |
| (1) ≤ 10 min (n = 203) | EI | INT-b (n = 23)* | INC-c (n = 8) |
| | n = 172 | EI predicted = 9 | EI predicted = 3 |
| (2) 10–30 min (n = 440) | INT-a (n = 226) | INT-c (n = 161) | INC-d (n = 53) |
| | EI predicted = 140 | EI predicted = 74 | EI predicted = 22 |
| (3) 30–60 min (n = 399) | INC-a (n = 69) | OA n = 660 | |
| | EI predicted = 41 | n = 194 | n = 136 |
| (4) >60 min (n = 367) | INC-b (n = 37) | n = 152 | n = 178 |
| | EI predicted = 17 | | |

Abbreviations: EI, extremely inactive category; OA, overtly active category; INT (a,b,c), physical activity levels intermediate between EI and OA; INC (a,b,c,d), incompatible physician and patient estimates of activity. (D)omestic, activities mainly confined to the home; (R)ecreational, predominantly outside the home; (A)ctive, predominantly devoted to maintaining fitness.

*EI predicted indicates the number of patients within each INT and INC subcategory reassigned to the EI category by the predictive algorithm.

of patients who were identified by the algorithm as having the EI phenotype (Table 1). A total of 21.7% of the full cohort (306/1409) were reassigned to EI. Of these, 15.8% (223/1409) were in the original INT a–c categories and 5.9% (83/1409) were in the original INC a–d categories. Thus, application of the algorithm increased the proportion of EI patients in the full cohort from 12.2% (172/1409) to 33.9% (478/1409).

Not surprisingly, comparisons of clinicopathological characteristics showed a trend towards worsening health status of patients in the order EI > INT > INC > OA (Table 2). The differences were particularly stark when comparing patients in the EI *versus* OA categories, while the INT group had intermediate values between the EI and OA groups. Fig 2 shows a comparison of selected anthropometric and behavioral characteristics (continuous variables) stratified by our PA categories or the GOLD ABCD 2017 categories. Of note, the symptom-related variables (mMRC, DIRECT, and CAT scores) logically discriminate between patients according to the GOLD ABCD classification, but they overlap the PA categories, indicating that these questionnaires individually have a poor ability to predict PA level. As shown in Fig 3, this possibility was confirmed by the large overlap between not only continuous variables (DIRECT score, CAT score, age, body mass index) but also categorical variables (age, sex, exacerbation, and GOLD ABCD) for patients in the EI, INT, INC, and OA categories, consistent with their poor individual ability to predict EI status.

## Predictive results

Table 3 shows the analysis of the predictive algorithm performance using several classifier methods. The BiMM and WRF results did not differ significantly, suggesting that the prediction was independent of the physician who collected the data and the practice setting. This assertion was further checked by performing a panel data analysis on the clustered data and testing the hypothesis of presence of random effects. This analysis yielded a p value of 0.0069, thus supporting a fixed effects model (i.e., a random forest prediction without random effects). Overall, the AUPRC indicates that the HyperSMURF algorithm achieved significantly better sensitivity than WRF or BiMM for predicting EI, with little deterioration in the sensitivity of the OA classification. As an example, Fig 3 shows the influence of some variable values on the prediction of EI status, and S3 Fig shows a comparable analysis for the prediction of OA. As

**Table 2. Clinical and functional characteristics of the stratified COPD patients (n = 1409).**

| | EI | INT | INC | OA | p-value |
|---|---|---|---|---|---|
| | n = 172 | n = 410 | n = 167 | n = 660 | |
| **Anthropometric and behavioral characteristics** | | | | | |
| Age (years) | 67.5 ± 10.1 | 65.4 ± 9.5 | 65.9 ± 8.6 | 65.5 ± 8.3 | 0.063 |
| Male gender | 60.5% | 63.9% | 61.7% | 73.5% | **** |
| BMI (kg/m$^2$) | 26.5 ± 6.8 | 26.2 ± 5.9 | 25.0 ± 5.4 | 25.8 ± 5.1 | 0.058 |
| Smokers (current or ex) | 0.965 | 0.963 | 0.964 | 0.964 | 0.07 |
| mMRC score | 2.7 ± 1.1 | 1.9 ± 1.1 | 1.8 ± 1 | 1.2 ± 0.9 | **** |
| DIRECT score | 17.4 ± 8.6 | 13.0 ± 8 | 12.0 ± 6.9 | 8.6 ± 6.4 | **** |
| CAT score | 21.3 ± 8.1 | 18.0 ± 7.6 | 17.4 ± 7.5 | 14.1 ± 7.2 | **** |
| HADS Anxiety subscore | 7.4 ± 4.6 | 6.3 ± 4.5 | 6.1 ± 4 | 5.4 ± 3.7 | **** |
| HADS Depression subscore | 8.2 ± 4.8 | 6.2 ± 4.2 | 5.6 ± 3.7 | 4.7 ± 3.5 | **** |
| **Functional respiratory parameters and GOLD 2011 classification** | | | | | |
| FEV$_1$ (L) | 1.28 ± 0.6 | 1.57 ± 0.6 | 1.58 ± 0.7 | 1.82 ± 0.7 | **** |
| FEV$_1$ (% predicted) | 50.9 ± 22.6 | 59.2 ± 22 | 59.2 ± 22.9 | 65.5 ± 20.5 | **** |
| FVC (L) | 2.5 ± 0.9 | 2.86 ± 0.9 | 3.0 ± 1.1 | 3.24 ± 1 | **** |
| FVC (% predicted) | 77.9 ± 24.3 | 85.4 ± 22.5 | 89.6 ± 25.2 | 92.8 ± 21.2 | **** |
| FEV$_1$/FVC (%) | 50.6 ± 14.5 | 54.4 ± 13.3 | 52 ± 14 | 55.3 ± 11.8 | **** |
| GOLD 1 | 13.4% | 18.3% | 21.6% | 24.7% | **** |
| GOLD 2 | 32.0% | 45.9% | 36.5% | 49.4% | **** |
| GOLD 3 | 25.6% | 24.4% | 29.3% | 21.7% | **** |
| GOLD 4 | 29.1% | 11.5% | 12.6% | 4.2% | **** |
| **Comorbidities and GOLD 2017 classification** | | | | | |
| Cardiovascular disease and/or diabetes | 83.1% | 71.0% | 69.5% | 63.9% | **** |
| Treated for anxiety or depression | 72.7% | 59.8% | 59.9% | 51.1% | **** |
| Exacerbation within the previous year (≥1 severe or ≥2 mild/moderate) | 48.3% | 32.7% | 39.5% | 25.0% | **** |
| GOLD A | 2.9% | 6.8% | 10.2% | 22.0% | **** |
| GOLD B | 48.8% | 60.5% | 50.3% | 53.0% | **** |
| GOLD C | 0.0% | 2.4% | 2.4% | 3.9% | **** |
| GOLD D | 48.3% | 30.2% | 37.1% | 21.1% | **** |

Data are presented as the percentage or mean ± standard deviation. Comparisons between PA categories were performed by Kruskal-Wallis tests and ANOVA with ordinal factors test (ordAOV). Significant differences are noted: $p < 0,...$****; $p < 0.001$ ***; $p < 0.01$ **; $p < 0.05$ *.

Abbreviations: BMI, body mass index; CAT, COPD Assessment Test; COPD, chronic obstructive pulmonary disease; DIRECT, Disability related to COPD Tool; FEV$_1$, forced expiratory volume in 1 s; FVC, forced vital capacity; GOLD, Global Initiative for Chronic Obstructive Lung Disease classification; HADS, Hospital Anxiety and Depression Scale; mMRC, modified Medical Research Council dyspnea scale. For EI, INT, INC, and OA definitions, see Table 1.

can be seen, only the higher scores (mMRC ≥3, CAT >30, DIRECT >23) are associated with a probability of EI >0.5. The strength of our predictive model is also confirmed by the corresponding ROC curves (Fig 4). Although the differences between the WRF and HyperSMURF predictions, as measured by the AUROC, are not large, AUPRC is considered to be more informative than AUROC for imbalanced data [24]. Finally, we applied our predictive algorithm process to the INT and INC subcategories. Table 1 shows that about half of the patients were predicted to be EI; specifically, 54% and 41% in the INT and INC categories, respectively.

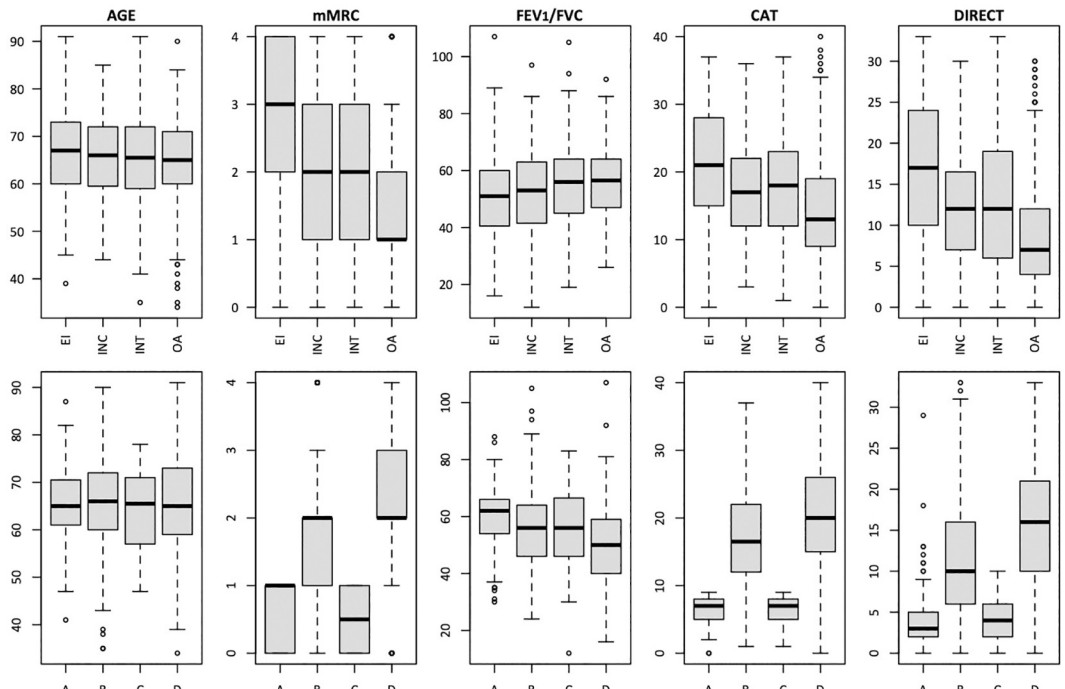

**Fig 2. Univariate boxplots comparing the distribution of selected continuous variables according to the physical activity category described here (top row) and GOLD 2017 category (bottom row).** Plots show the median, minimum, maximum, and interquartile values. See Table 1 for definitions of activity categories.

S1 Table shows the distribution of the GOLD 2011 and ABCD classifications within the PA categories.

## Discussion

The main contribution of this study is to demonstrate the predictive validity of an algorithm for predicting the least active COPD patients from information available in routine pulmonologist practice independently of PA-related measures. The originality and strength of our algorithm lies in its ability to predict EI in patients whose PA level is equivocal or unclear based on the patient's and physician's opinions, thus bringing to light the precise subgroup of COPD patients who are most in need of increased PA. Depending on the options available to the referring pulmonologist, this algorithm will help in deciding the optimal next step for each patient; whether that is accelerometry, as proposed by the PROactive consortium,[8] referral to supervised rehabilitation [25], and/or simply encouraging the patient to participate in social activities that include PA [26].

### Selection of machine learning methods

In the present study, we demonstrate that a specific random forest machine learning algorithm, which we refer to as the EI algorithm, is effective in predicting the EI or OA status of COPD patients. In addition, the algorithm has the potential to automatically detect the most informative predictors of EI by excluding many irrelevant confounding factors that influence both the dependent variable (EI or OA) and independent variables (explanatory variable), thus causing a spurious association. The EI algorithm outperforms traditional multiple linear/logistic regression models by unmasking predictive potential not apparent in a linear model. We

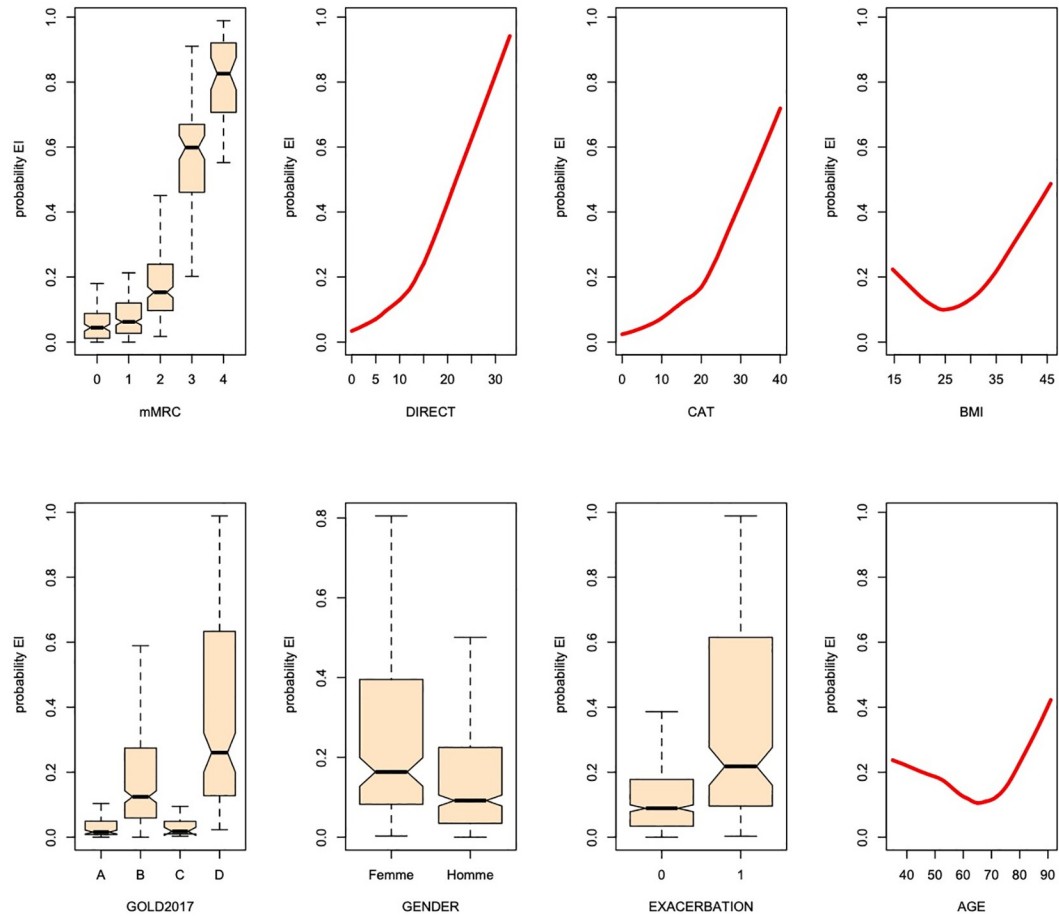

**Fig 3. Box plots (categorical/ordinal variables) and line plots (continuous variables) of the marginal effect of a predictor (x-axis) on the probability of a patient being assigned to the EI category according to the weighted random forest method (y-axis).** See also S3 Fig for the inverse analysis of probability of assignment to the OA category. Box plots show the median, minimum, maximum, and interquartile values. See Table 1 for definitions of activity categories.

could also have considered using a Bayesian machine learning framework to develop a prediction procedure and simultaneously identify promising subsets of relevant predictors. While the Bayesian framework may have achieved equivalent predictive performance, it would have

**Table 3. Evaluation of the performance of the predictive algorithm.**

| | Overall error | Accuracy* | PPV | NPV | Sensitivity | | AUPRC | AUROC* |
|---|---|---|---|---|---|---|---|---|
| | | | | | EI | OA | | |
| HyperSMURF | 13.7% | 0.76 (0.69–0.82) | 0.45 | 0.93 | 79.4% | 75.0% | 0.64 | 0.84 (0.75–0.92) |
| Weight Random Forest | 14.1% | 0.84 (0.87–0.90) | 0.63 | 0.90 | 59.0% | 90.9% | 0.49 | 0.75 (0.66–0.84) |
| BiMM Random Forest | 14.2% | 0.84 (0.78–0.90) | 0.87 | 0.67 | 47.0% | 94.0% | 0.47 | 0.70 (0.62–0.80) |

*Accuracy and AUROC are presented with 95% confidence intervals.

Data are based on analysis of 20% (n = 166) patients in the OA and EI groups.

Abbreviations: AUROC, area under the receiver operating characteristic curve; AUPRC, area under the precision and recall curve; BiMM, binary mixed model forest algorithm (23); HyperSMURF, hyper-SMOTE under sampled random forests (24); NPV, negative predictive value; PPV, positive predictive value. See Table 1 for definitions of EI and OA.

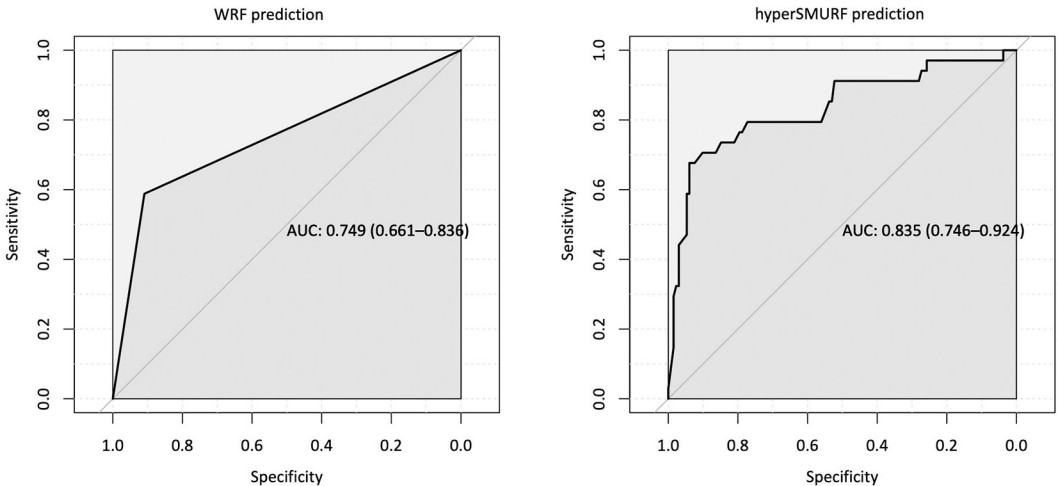

**Fig 4.** Receiver operating characteristic curves for the prediction of EI using weighted random forest (WRF, left) and hyper-ensemble of SMOTE under sampled random forests (HyperSMURF, right) methods. Areas under the curves are shown as the median and 95% confidence intervals.

required a large number of assumptions on independent variables and many successive statistical checks, making it much more difficult to interpret. Because of this complexity, we opted for a frequentist framework that markedly reduces the number of mathematical steps and their validation and obtains a level of predictive validity acceptable for its intended clinical use.

Our results confirm that the EI algorithm possesses two critical features of a predictive model: the agreement between observed probabilities and predicted probabilities (i.e., calibration) and the ability to clearly distinguish between categories (i.e., discrimination). Thus, for the intended purpose of guidance in clinical decision-making, our EI algorithm provides an acceptable balance between a high rate of true positives (correctly identified patients) and a low rate of false positives (incorrectly identified patients). As with any predictive algorithm designed to assist in medical decisions, the EI algorithm should be considered a contributing tool that takes into account the potential impact on the patient's health.

### Decision-making process and machine learning

Matching these predicted probabilities with a 0–1 classification, by choosing a threshold above which a new observation is classified as 1 *versus* 0, is no longer part of the statistics. It is part of the decision-making process that integrates other contingencies or issues than the probabilistic results of the model. Practitioners may ask several pertinent questions that could influence this threshold. For example, will a binary categorization (EI and OA) negatively affect patient care compared with a more detailed determination of daily PA behavior? If so, in what way will it affect care, especially with respect to the design of individualized pulmonary rehabilitation programs or personalized recommendations? Like any diagnostic method implemented in the clinical decision-making process, the predictive validity of the information and the operational impact of the level of precision must be evaluated. This echoes some points raised by Faner and Agusti [27], who questioned the practical use of conclusions based on clustering studies for identifying a clinical phenotype predictive of mortality for a single patient. In that case, the issue was whether a complex analytical approach—as opposed to common sense—was really necessary to know that patients with severe airflow limitation and comorbidities would have a poor prognosis.

In real-life practice, the purpose of the EI algorithm would be to alert the physician to the probability of a new patient having EI or OA status. This is particularly important because only a minority of patients who are eligible for pulmonary rehabilitation actually derive benefit [28], partly because the referring pulmonologist may be unaware of the patient's true EI status, which may be sufficiently poor as to predispose them to failure. In support of this, our results suggest that the most extreme inactivity (i.e., EI) is largely underestimated in routine consultations. Indeed, application of the EI algorithm increased the proportion of the total population with EI status from 12.2%, detected by the patient and physician estimates, to 21.7%. Our results compare favorably with those reported by Schneider et al. [5], who examined daily PA in COPD patients using accelerometry. The detailed analysis of the kinetics and intensity of PA by those authors found that 49% (n = 67) of patients could be defined as "active and non-sedentary" and 26% (n = 35) as "non-active and sedentary", which compare with 46.8% OA (n = 660) and 34% EI (n = 478) in our study. Nevertheless, further comparisons between studies based on accelerometry measurements and machine learning using non-PA data are beyond the scope of this analysis.

## Limitations and strengths

The method we have proposed to define EI status may seem too simplistic compared with objective measurements from accelerometry. Our definition was based on two assumptions: that employing both patient- and physician-derived information would compensate for any imprecision resulting from subjectivity; and that EI status could be predicted from routine clinical data (e.g., behavioral, psychological, symptomatic) that are causes and/or consequences of extreme inactivity. It is important to note that whether the EI status used here would be exactly the same as one derived from accelerometry is ultimately not a crucial factor.

The most important intended use of the algorithm is to enable patients with genuine EI status to be identified when the clinical data are equivocal. The best illustration that our assumptions were acceptable is the accuracy of prediction with the test sample of EI and OA patients (n = 166), which had a modest predictive error of 13.7%. Another limitation is that we did not perform accelerometry of the 306 patients with intermediate PA levels who were reclassified by our algorithm as EI. However, various studies have reported that between 10% and 20% of data are routinely missing from accelerometry studies (incomplete measurements or any other reasons) and the patient number included per study rarely exceeds about 100. In addition, considering that >200 pulmonologists from throughout France contributed data to the EI algorithm, any attempt to perform comparative accelerometry would undoubtedly have resulted in an even higher rate of lost or unusable data. We propose that the predictive validity of our predictive algorithm will increase as the size and diversity of the COLIBRI-COPD database increases. Moreover, the addition of new variables to the EI algorithm is technically possible, because the machine learning approach developed for the algorithm is an evolutionary and adaptable process. Examples of potentially influential variables for predicting EI status are psychological and social vulnerability, and regional climate and pollution data [9]. The addition of physiological data, such as functional exercise capacity (6-minute walk test, chair-rising test, grip strength, pedometer readings) could also be valuable, even though these parameters have been shown to be individually unreliable for identifying patients with extremely inactive lifestyles [11].

## Interpretation

In conclusion, we report that a predictive machine learning algorithm, developed from routine clinical data collected during online consultations, can identify EI status among patients with

all stages of COPD severity. Integration of this algorithm within online consultations *via* an R-Shiny-python interface [29] could alert the clinician to the frequently overlooked patients who urgently require intervention to promote PA. Thus, it is our hope that the approach proposed here will advance the field of medical decision-making and move it further towards the holy grail of predictive and personalized medicine for COPD patients.

## Supporting information

**S1 Fig. 2D plot of the first two canonical discriminant variables accounting for the greatest variation between physical activity categories (red) relative to error.** The two dimensions account for 98.6% of the variance between categories, most (95.8%) of which is due to EI *versus* OA. The latter is mainly influenced by $FEV_1/FVC$ and the former by CAT, DIRECT, and mMRC scores.
(TIFF)

**S2 Fig. Schematic representation of the HyperSMURF method.** HyperSMURF divides the majority class (OA) into n partitions. For each partition, oversampling techniques are used to generate additional patients from the minority class (EI) that closely resemble the distribution of the actual class to amplify the number of training patients from the minority class. At the same time, a comparable number of patients is subsampled from the majority class. HyperSMURF then trains in parallel n random forests on the resulting balanced data sets and finally combines the prediction of the n ensembles according to a hyper-ensemble (ensemble of ensembles) approach.
(TIFF)

**S3 Fig. Box plots (categorical/ordinal values) and line plots (continuous variables) of the marginal effect of a predictor (x-axis) on predicted probability of a patient being assigned to the OA category according to the weighted random forest method (y-axis).** Box plots show the median, minimum, maximum, and interquartile values. See Table 1 for definitions of activity categories.
(TIFF)

**S1 Table. Distribution of patients classified as GOLD ABCD within the INT and INC physical activity categories.**
(PDF)

## Acknowledgments

We thank Pr. François Peronnet for his critical review and Anne M. O'Rourke, PhD, for editing a draft of the manuscript.

## Author Contributions

**Conceptualization:** Bernard Aguilaniu, Anestis Antoniadis.

**Data curation:** David Hess, Pei Zhi Li, Anestis Antoniadis.

**Formal analysis:** Bernard Aguilaniu, David Hess, Pei Zhi Li, Anestis Antoniadis.

**Funding acquisition:** Bernard Aguilaniu, David Hess.

**Investigation:** Bernard Aguilaniu, Eric Kelkel, Amandine Briault, Marie Destors, Jacques Boutros.

**Methodology:** Bernard Aguilaniu, Pei Zhi Li, Anestis Antoniadis.

**Project administration:** David Hess, Eric Kelkel.

**Resources:** David Hess.

**Software:** David Hess.

**Supervision:** Anestis Antoniadis.

**Validation:** Bernard Aguilaniu, Anestis Antoniadis.

**Writing – original draft:** Bernard Aguilaniu, Anestis Antoniadis.

**Writing – review & editing:** Bernard Aguilaniu, David Hess, Eric Kelkel, Amandine Briault, Marie Destors, Jacques Boutros, Anestis Antoniadis.

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
