## [Decision Letter · Decision Letter 0]

29 Apr 2021

PONE-D-21-07345

A machine learning approach to predict extreme inactivity in COPD patients using non-activity-related clinical data

PLOS ONE

Dear Dr. Aguilaniu,

Thank you for submitting your manuscript to PLOS ONE. After careful consideration, we feel that it has merit but does not fully meet PLOS ONE’s publication criteria as it currently stands. Therefore, we invite you to submit a revised version of the manuscript that addresses the points raised during the review process.

We look forward to receiving your revised manuscript.

Kind regards,

Prof. Jeremy Coquart, Ph.D.

Academic Editor

PLOS ONE

Journal Requirements:

2. Please amend your Methods section to add the ethics information and approval numbers that you provided in your Ethics Statement.

[The COLIBRI web consultation platform is supported by contractual partnerships with Agir à Dom, AstraZeneca, Boehringer Ingelheim, Chiesi, GlaxoSmithKline and Novartis.

BA, DH, and AA received grants from Agir à Dom, AstraZeneca, Boehringer Ingelheim, Chiesi, GlaxoSmithKline, and Novartis for the conduct of the study.

The funders had no role in study design, data collection and analysis, decision to publish, or preparation of the manuscript.]. 

We note that you received funding from a commercial source: Agir à Dom, AstraZeneca, Boehringer Ingelheim, Chiesi, GlaxoSmithKline and Novartis

5. One of the noted authors is a group or consortium [COLIBRI-COPD program contributors]. In addition to naming the author group, please list the individual authors and affiliations within this group in the acknowledgments section of your manuscript. Please also indicate clearly a lead author for this group along with a contact email address.

6. Please amend the manuscript submission data (via Edit Submission) to include author 'COLIBRI-COPD program contributors'.

Reviewers' comments:

Reviewer's Responses to Questions

**Comments to the Author**

1. Is the manuscript technically sound, and do the data support the conclusions?

Reviewer #1: Partly

2. Has the statistical analysis been performed appropriately and rigorously? 

Reviewer #1: Yes

3. Have the authors made all data underlying the findings in their manuscript fully available?

Reviewer #1: No

4. Is the manuscript presented in an intelligible fashion and written in standard English?

Reviewer #1: Yes

5. Review Comments to the Author

Reviewer #1: Abstract:

The rational between the first and second sentences is missing. In other words, why would one predict physical activity level from clinical data using artificial intelligence?

Why the algorithm was proposed to identify IE patients in the INT and INC groups is not clear to me. I don't understand how the authors can be sure that the patients have been classified correctly.

Introduction:

This part is pretty well written. It just lacks, in my opinion, a paragraph on artificial intelligence... because the authors could have proposed simple linear equations to classify their patients. However, they use machine learning.

Then, in my opinion, the end of the introduction (from "Since 2013, pulmonologists...") should be in the "method" section instead. The authors indicate what they have done.

Method:

For clarity, the authors could list each of the 22 specific variables in parentheses. For example, "...in the areas of anthropometry (age, body mass index...).

Results:

The authors do not present significant differences within groups (p values). This could be added especially for Table 2. This is especially important because the authors state: “Not surprisingly, comparisons of clinicopathological characteristics showed a trend towards worsening health status of patients in the order EI > INT > INC > OA (Table 2).”

The final equation (with the weight of each variable: age, BMI, FEV1…) must be presented. Without it, physicians cannot classify their patients, and therefore the study loses its interest. In other words, readers should be able to access the equation that will allow them to tell if their patients are extremely inactive or not...

Discussion:

In the first sentence of the discussion, the authors say that they have shown the reliability of their method. To me, it does not show that, but rather the validity.

As mentioned before, the authors state here that "Depending on the options available to the referring pulmonologist, this algorithm will help in deciding the optimal next step for each patient..." but the authors do not give their algorithm. This should be presented in the article.

The authors are aware of the limitations of their study (mainly self-reported information, subjectivity for ranking, lack of GPS measurement...). However, there is another limitation. Indeed, 800 patients is a lot, but not much for machine learning... Often to gain accuracy (actually only 86%), several thousands of data are needed…

Again, the authors talk about reliability, but for me it does not test that.

6. PLOS authors have the option to publish the peer review history of their article (what does this mean?). If published, this will include your full peer review and any attached files.

Reviewer #1: No

---

## [Author Response · Author response to Decision Letter 0]

5 Jul 2021

B. Abstract:

Reviewer: The rational between the first and second sentences is missing. In other words, why would one predict physical activity level from clinical data using artificial intelligence?

Extreme inactivity contributes to mortality and morbidity in patients with chronic obstructive pulmonary disease (COPD), but patient-reported physical activity (PA) levels are often inaccurate. Here, we describe a machine learning algorithm that uses non-PA-related clinical data to identify extreme inactivity in patients with equivocal PA status.

Answer: Indeed, an intermediate sentence is missing. Thank you for the relevance of this observation. We propose a new text.

Facilitating the identification of extreme inactivity (EI) has the potential to improve morbidity and mortality in COPD patients. Apart from patients with obvious EI, the identification of a such behavior during a real-life consultation is unreliable. We therefore describe a machine learning algorithm to screen for EI, as actimetric measurements are difficult to implement.

Reviewer: Why the algorithm was proposed to identify IE patients in the INT and INC groups is not clear to me. 

Answer: The previous sentence that has been corrected makes it more explicit why the ML algorithm seeks to identify EI patients among those whose EI behavior is not clinically evident (INT and INC). Thanks again to the reviewer for this remark.

 We therefore describe a machine learning algorithm to screen for EI, as actimetric measurements are difficult to implement

Reviewer: I don't understand how the authors can be sure that the patients have been classified correctly.

Answer: The answer to this question is discussed in more detail below when we discuss the difference between reliability and predictive validity.

C. Introduction:

Reviewer: This part is pretty well written. It just lacks, in my opinion, a paragraph on artificial intelligence

Answer: Due to the limited number of words allowed and also because we have clearly detailed the steps in the development of the AI algorithm, we have not found it useful to explain the AI process further

Reviewer: ... because the authors could have proposed simple linear equations to classify their patients. However, they use machine learning.

Answer: We have indeed approached the problem more classically with multiple logistic regression techniques for binary variables but with less powerful results. Our choice was also motivated by the literature of which we report below 2 examples which compare the 2 predictive approaches. 

References:

1. Kaitlin Kirasich, Trace Smith and Bivin Sadler (2018). Random Forest vs Logistic Regression: Binary Classification for Heterogeneous Datasets. SMU Data Science Review, Volume 1, Number 3, Article 9.

2. Couronné, R., Probst, P. & Boulesteix, AL. Random forest versus logistic regression: a large-scale benchmark experiment. BMC Bioinformatics 19, 270 (2018). https://doi.org/10.1186/s12859-018-2264-5

Reviewer: Then, in my opinion, the end of the introduction (from "Since 2013, pulmonologists...") should be in the "method" section instead. The authors indicate what they have done.

Answer: Indeed, we moved these last sentences (from “Since 2013…”) in the method section opening a short paragraph “Construct of the predictive machine learning”

D. Methods:

Reviewer: For clarity, the authors could list each of the 22 specific variables in parentheses. For example, "...in the areas of anthropometry (age, body mass index...).

Answer: In accordance with the reviewer's recommendation, we specified that the 22 variables considered were presented in Table 2

We selected 1409 patients with comprehensive information on 22 specific variables (see table 2) in the areas of anthropometry, smoking habits, resting pulmonary function, comorbidities, exacerbations during the preceding year, Global Initiative for COPD (GOLD) ABCD classification, and self-reported questionnaires: the modified Medical Research Council dyspnea scale (mMRC) [16] and Disability related to COPD Tool (DIRECT), both of which assess dyspnea [17, 18] ; the COPD Assessment Test (CAT), which assesses quality of life [19] ; and the Hospital Anxiety and Depression Scale, which separately assesses anxiety and depression [20, 21]. 

E. Results:

Reviewer: The authors do not present significant differences within groups (p values). This could be added especially for Table 2. This is especially important because the author’s state: “Not surprisingly, comparisons of clinicopathological characteristics showed a trend towards worsening health status of patients in the order EI > INT > INC > OA (Table 2).”

Answer: Classical one-way ANOVA can be seen as a generalization of the t-test for comparing the means of a continuous variable in more than two groups defined by the levels of a discrete covariate, a so-called nominal factor. Testing is then typically done by using the standard F-test.

However, when it comes to use an ordinal factor (factor's levels are ordered), the choices for performing an appropriate ANOVA analysis are slim. An alternative test to the classical F-test, taking the ordering of factor levels into account has been developed in [1] using a mixed model’s methodology. Software implementing the proposed ANOVA procedure for factors with ordered levels in the above paper is included in R package ``ordPens'' (see [2]). This function performs analysis of variance when the factor(s) of interest has/have ordinal scale level. For testing, values from the null distribution are simulated. The method uses a mixed effects formulation of the usual one- or multi-factorial ANOVA model (with main effects only) while penalizing (squared) differences of adjacent means. The interested reviewer is referred to the above-mentioned paper for further details.

1. Gertheiss, J. (2014). ANOVA for Factors With Ordered Levels. Journal of Agricultural, Biological, and Environmental Statistics, Volume 19, Number 2, Pages 258–277. https://doi.org/10.1007/s13253-014-0170-5

2. Gertheiss J. (2015) ordPens: Selection and/or Smoothing of Ordinal Predictors. R package version 0.3-1 (2015). Available online at: https://CRAN.R-project.org/package=ordPens

Reviewer: The final equation (with the weight of each variable: age, BMI, FEV1…) must be presented. Without it, physicians cannot classify their patients, and therefore the study loses its interest. In other words, readers should be able to access the equation that will allow them to tell if their patients are extremely inactive or not...

Answer 1: We do not make logistic regressions but Random Forests. Therefore, there is no equation formula to make the prediction (see references above). The prediction is made from the fitted model which also has the advantage of being able to increase its predictive power by introducing new variables such as a functional capacity test (for example: sit-to-stand or 6 min walking tests, etc.). This point has been in the discussion section:

Moreover, the addition of new variables to the EI algorithm is technically possible, because the machine learning approach developed for the algorithm is an evolutionary and adaptable process….

Answer 2 : It is indeed essential that readers have access to the ML. As stated in the conclusion, the return of an individual result is done by a site to which the instantaneous calculation process is attached.

Integration of this algorithm within online consultations via an R-Shiny-python interface29 could alert the clinician to the frequently overlooked patients who urgently require intervention to promote PA

F. Discussion:

Reviewer: In the first sentence of the discussion, the authors say that they have shown the reliability of their method. To me, it does not show that, but rather the validity.

Answer: We thank reviewer for this pertinent comment. In the context of predictive statistics, it is indeed better to use the term predictive validity than reliability. Therefore, we modify the text in this sense. In addition, we provide below additional comments (which we do not offer to readers because it deals specifically with statistical considerations remote from medical interests) to convince that the reliability of the statistical model is robust. In addition, we provide below additional elements to convince the reviewer that the reliability of the statistical model is robust; in other words that the predictive validity is very satisfactory.

The predictive models developed in our paper have been evaluated in terms of predictive reliability (discrimination), i.e. the ability to differentiate between high and low risk events, and calibration, or the accuracy of the risk estimates. An accurate probability estimate is crucial for clinical decision making and well-calibrated predictive models are imperative in our case. Unfortunately, even a highly discriminative classifier (e.g., a classifier with a large area under the receiver operating characteristic (ROC) curve, or AUROC) may not be well-calibrated. Various techniques have been proposed to calibrate existing predictive models. We have used in our case the GiViTI calibration belt and associated test which apply to models estimating the probability of binary responses, such as the random forest regression models studied in our paper (see references [1] and [2]). In particular, we have adopted the approach implemented in the R-package GiViTI which allows to evaluate the models’ external calibration, in independent samples (test) different than the training dataset used to fit the model. 

Calibration belt and test: 

The GiViTI calibration belt has used to assess the calibration of the fitted Random Forest model. External validation has been be applied on the testing part of the data. The function givitiCalibrationBelt generates the plot displayed in the figure below.

By default, the 80%- and 95%-confidence level calibration belt are plotted, in light and dark grey respectively. The table in the bottom-right side of the figure reports the ranges of the predicted probabilities where the belt significantly deviates from the bisector. Notably, the calibration belt contains the bisector (representing the identity between predicted probability and observed response rate) for all predictions in our RF fit. Hence, the RF predictions match the average observed rates in the whole range without any overestimation of the risk of EI or OA patients. The overall calibration of the model is synthesized into the test’s p-value, which is reported in the top-left corner of the figure. In addition, the sample size \\(n=166\\) of the external (test) sample and the polynomial order of the calibration curve (for an explanation see reference [1]) are reported in the plot. The conclusion is that the RF model that was developed is well calibrated and doesn't overestimates the probability of the risk events. 

[1]. Finazzi, S., D. Poole, D. Luciani, P. E. Cogo, and G. Bertolini. 2011. “Calibration Belt for Quality of Care Assessment Based on Dichotomous Outcome.” PLoS ONE 6 (2): e16110. doi:10.1371/journal.pone.0016110.

[2]. Nattino, Giovanni, Stefano Finazzi, and Guido Bertolini. 2014. “A New Calibration Test and a Reappraisal of the Calibration Belt for the Assessment of Prediction Models Based on Dichotomous Outcomes.” Statistics in Medicine. doi:10.1002/sim.6100.

Reviewer: As mentioned before, the authors state here that "Depending on the options available to the referring pulmonologist, this algorithm will help in deciding the optimal next step for each patient..." but the authors do not give their algorithm. This should be presented in the article.

Answer: As previously mentioned, the ML algorithm delivers the predictive result via a web-based application that instantly calculates the prediction of being or not being EI. This application will be available upon final acceptance of the article.

Reviewer: The authors are aware of the limitations of their study (mainly self-reported information, subjectivity for ranking, lack of GPS measurement...). However, there is another limitation. Indeed, 800 patients is a lot, but not much for machine learning... Often to gain accuracy (actually only 86%), several thousands of data are needed…

Answer: We are fully aware of the remarks raised by the reviewer. But as it is underlined in the article and in the response to the comments above, our ML is designed to improve the predictive performance as the number of patients included in the Colibri-COPD cohort increases but also by enrichment with new variables of interest

Reviewer: Again, the authors talk about reliability, but for me it does not test that.

See answer above.

---

## [Decision Letter · Decision Letter 1]

28 Jul 2021

A machine learning approach to predict extreme inactivity in COPD patients using non-activity-related clinical data

PONE-D-21-07345R1

Dear Dr. Aguilaniu,

We’re pleased to inform you that your manuscript has been judged scientifically suitable for publication and will be formally accepted for publication once it meets all outstanding technical requirements.

Kind regards,

Prof. Jeremy Coquart, Ph.D.

Academic Editor

PLOS ONE

Additional Editor Comments (optional):

Reviewers' comments:

Reviewer's Responses to Questions

**Comments to the Author**

1. If the authors have adequately addressed your comments raised in a previous round of review and you feel that this manuscript is now acceptable for publication, you may indicate that here to bypass the “Comments to the Author” section, enter your conflict of interest statement in the “Confidential to Editor” section, and submit your "Accept" recommendation.

Reviewer #1: All comments have been addressed

2. Is the manuscript technically sound, and do the data support the conclusions?

Reviewer #1: Yes

3. Has the statistical analysis been performed appropriately and rigorously? 

Reviewer #1: Yes

4. Have the authors made all data underlying the findings in their manuscript fully available?

Reviewer #1: Yes

5. Is the manuscript presented in an intelligible fashion and written in standard English?

Reviewer #1: Yes

6. Review Comments to the Author

Reviewer #1: Thanks to the authors for answering all my questions. Even if all my remarks were not included, the paper could be published now.

7. PLOS authors have the option to publish the peer review history of their article (what does this mean?). If published, this will include your full peer review and any attached files.

Reviewer #1: No

---

## [Editor Report · Acceptance letter]

11 Aug 2021

PONE-D-21-07345R1 

A machine learning approach to predict extreme inactivity in COPD patients
using non-activity-related clinical data 

Dear Dr. Aguilaniu:

I'm pleased to inform you that your manuscript has been deemed suitable for publication in PLOS ONE. Congratulations! Your manuscript is now with our production department. 

Kind regards, 

on behalf of

Professor Jeremy Coquart 

Academic Editor

PLOS ONE